# Development and Functionalization of a Novel Chitosan-Based Nanosystem for Enhanced Drug Delivery

**DOI:** 10.3390/jfb14110538

**Published:** 2023-11-01

**Authors:** Carmen Grierosu, Gabriela Calin, Daniela Damir, Constantin Marcu, Radu Cernei, Georgeta Zegan, Daniela Anistoroaei, Mihaela Moscu, Elena Mihaela Carausu, Letitia Doina Duceac, Marius Gabriel Dabija, Geta Mitrea, Cristian Gutu, Elena Roxana Bogdan Goroftei, Lucian Eva

**Affiliations:** 1Faculty of Dental Medicine, “Apollonia” University of Iasi, 11 Pacurari Str., 700511 Iasi, Romania; grierosucarmen@yahoo.com (C.G.); letimedr@yahoo.com (L.D.D.); lucianeva74@yahoo.com (L.E.); 2Orthopaedic Trauma Surgery Clinic, Clinical Rehabilitation Hospital, 14 Pantelimon Halipa Str., 700661 Iasi, Romania; 3Faculty of Medicine, “Grigore T. Popa” University of Medicine and Pharmacy Iasi, 16 Universitatii Str., 700115 Iasi, Romania; 4Faculty of Medicine and Pharmacy, University Dunarea de Jos, 47 Domneasca Str., 800008 Galati, Romania; marcu_saar@yahoo.de (C.M.); getamitrea@yahoo.com (G.M.); dr_cgutu@yahoo.com (C.G.); elenamed84@yahoo.com (E.R.B.G.); 5Saarbrucken-Caritas Klinkum St. Theresia University Hospital, 66113 Saarbrücken, Germany; 6Faculty of Dental Medicine, “Grigore T. Popa” University of Medicine and Pharmacy Iasi, 16 Universitatii Str., 700115 Iasi, Romania; georgetazegan@yahoo.com (G.Z.); anistoroaei_daniela@yahoo.com (D.A.); mihaela_moscu@ymail.com (M.M.); mihaelacarausu@yahoo.com (E.M.C.); mariusdabija.md@gmail.com (M.G.D.); 7“Prof. Dr. Nicolae Oblu” Neurosurgery Hospital Iasi, 2 Ateneului Str., 700309 Iasi, Romania; 8“St. Ap. Andrei” Emergency Clinical Hospital, 177 Brailei Str., 800578 Galati, Romania; 9“Dr. Aristide Serfioti” Emergency Military Hospital, 199 Traian Str., 800150 Galati, Romania; 10“Sf Ioan” Emergency Clinical Hospital, 2 Gheorghe Asachi Str., 800494 Galati, Romania

**Keywords:** polymer composite, bioactivity, biodegradability, metronidazole, biomedical application

## Abstract

Nowadays, infection diseases are one of the most significant threats to humans all around the world. An encouraging strategy for solving this issue and fighting resistant microorganisms is to develop drug carriers for a prolonged release of the antibiotic to the target site. The purpose of this work was to obtain metronidazole-encapsulated chitosan nanoparticles using an ion gelation route and to evaluate their properties. Due to the advantages of the ionic gelation method, the synthesized polymeric nanoparticles can be applied in various fields, especially pharmaceutical and medical. Loading capacity and encapsulation efficiency varFied depending on the amount of antibiotic in each formulation. Physicochemical characterization using scanning electron microscopy revealed a narrow particle size distribution where 90% of chitosan particles were 163.7 nm in size and chitosan-loaded metronidazole nanoparticles were 201.3 nm in size, with a zeta potential value of 36.5 mV. IR spectra revealed characteristic peaks of the drug and polymer nanoparticles. Cell viability assessment revealed that samples have no significant impact on tested cells. Release analysis showed that metronidazole was released from the chitosan matrix for 24 h in a prolonged course, implying that antibiotic-encapsulated polymer nanostructures are a promising drug delivery system to prevent or to treat various diseases. It is desirable to obtain new formulations based on drugs encapsulated in nanoparticles through different preparation methods, with reduced cytotoxic potential, in order to improve the therapeutic effect through sustained and prolonged release mechanisms of the drug correlated with the reduction of adverse effects.

## 1. Introduction

For many years, nanotechnology significantly contributed to biomedicine regarding prevention, diagnosis and treatment of various diseases [1,2,3]. Nanomaterials used as drug transporters are very important due to their small size and morphology [4,5]. These properties are relevant for active molecules loading and sustained release to the specific site of action [6,7,8,9]. A considerable issue is to design drug delivery systems in order to enhance the pharmaceutical impact of a drug and to limit its side effects [10,11].

Metronidazole (MET) is a broad-spectrum antibiotic used to combat anaerobic bacteria and some parasites. It is a nitro-imidazole (Figure 1) derived from the reduction of the nitro group on the molecule by the bacteria and thus determines the emergence of toxic metabolites. These compounds use their bactericidal action with molecular DNA destruction, stopping the DNA repair process. Metronidazole is a drug that penetrates bacteria through the mechanism of passive diffusion, and is activated in the cytoplasm where it is transformed into free nitrogen radical. Having cytotoxic activity, it inhibits synthesis and damages DNA, stopping the multiplication of bacteria. Furthermore, the DNA breakage caused by the metronidazole metabolites leads to bacterial cell damage. This drug treats surgical infections, duodenal ulcer caused by Helicobacter Pylori contamination, intestinal amoebiasis, etc. Nowadays, metronidazole is used as an antimicrobial agent for the cure of periodontal disease, including topical application after scaling and subgingival treatment. The pharmacokinetic profile of metronidazole shows that the active substance reaches a concentration of 10 mg/mL in plasma one hour after the administration of a dose of 500 mg.

Administration of MET can cause certain side effects such as nausea, mouth dryness, epigastric pain and others. A major disadvantage regarding conventional therapy refers to the accumulation of high drug concentrations in the liver and kidneys. Among the side effects caused by the administration of systemic antibiotics, gastrointestinal intolerances, hypersensitivity reactions and the establishment of bacterial resistance can be noted. There are studies that reveal that if the active substance does not reach the desired concentration at the site of action, systems for the prolonged release of drugs can be developed, reaching the desired concentration at the site of action and reducing adverse effects.

In order to combat adverse effects of metronidazole it is necessary to reformulate the antibiotic by developing a different drug delivery nanosystem which yields better targeted transport of the active molecule. Nowadays, researchers are focused on reducing antibiotic side effects as well as delivering the active molecule to the target site, enhancing drug efficiency [12,13,14,15,16]. MET is also currently used for the treatment of periodontal disease, especially as a topical administrative drug inhibiting anaerobic microorganisms [17,18].

Although many types of nanoparticles were studied as drug delivery systems, ample researches were developed on polymeric nanostructures for various active molecule transports. Polymeric nanosized architectures own excellent stability, reproducibility and biodegradability properties, making them suitable for the application in several pharmaceutical formulations [16].

Chitosan is a natural polymer obtained with the deacetylation of chitin. It is a biocompatible cationic polysaccharide metabolized with specific human enzymes, mainly lysozyme, which attributes chitosan to be a possibly efficient drug carrier [19,20,21]. Chemical structure of chitosan (Figure 2) contains copolymer units of N-acetyl-glucosamine and glucosamine. Structurally, the chitosan molecule presents functional groups that allow electrochemical interactions at the molecular and cellular level. Chitosan nanoparticles possess antimicrobial properties; therefore, they can be loaded with several bioactive molecules to obtain different formulations used in the medical field [22,23,24,25]. Due to chitosan’s low solubility, some proposals for tailoring its structure were advanced in order to obtain various chitosan derivatives, and improved solubility was achieved by using free hydroxyl and amino groups based on the self-assembly feature.

The significant properties of chitosan determined researchers to use it in various bio-medical formulations. 

Studies on the synthesis of nanoparticles with the ionic gelation method have attracted the attention of researchers in recent years. This is because various compounds can be loaded into the polymer structure, thus constituting the drug delivery system and being able to be used later in various medical fields. Therefore, this biopolymer has been involved in several trials, namely drug delivery, with applications in many medical areas, such as dental medicine, tissue engineering, epidemiology, pulmonology, neurosurgery, neonatology, cardiology, emergency, regenerative medicine (including hard tissue and soft tissue due to its ability to protect unstable biomolecules), biodegradation, biocompatibility, mucosal tissue adhering, nontoxicity and antimicrobial activity. The use of such systems requires small amounts of the active substance as the absorption of the medicine is carried out in a sustained and controlled manner, thus reducing the side effects of antibiotics [26,27,28,29,30,31]. Although many drug-loaded chitosan nanoparticles were studied as carriers, this work proposes obtaining a novel formulation based on metronidazole-encapsulated chitosan nanoparticles and evaluating the cytotoxicity of these nanostructures according to physicochemical features and the drug release profile. Nowadays, biomedical application and development of polymer composites [32], which includes significant features such as bioactivity, low toxicity, molecules transport and biodegradability, is of major interest. Our study involved the preparation of a nanosystem based on chitosan nanoparticles loaded with metronidazole, which was structurally and morphologically characterized and evaluated for its cytotoxic effect. Formulation designs based on drug-loaded polymeric nanoparticles could provide improved therapeutic alternatives for currently administered drugs. Furthermore, this study proposed that the new drug delivery systems increase the therapeutic effect through biodegradation processes of the polymer matrix, deliver the active molecule to the target site and, implicitly, reduce side effects on the body.

## 2. Materials and Methods

### 2.1. Materials

Metronidazole active substance was offered by a research institute for free (Sigma–Aldrich, analytical standard, ≤100%, stable at room temperature). Chitosan (75–85% de-acetylated) and sodium tri-polyphosphate was acquired from Sigma–Aldrich, Darmstadt, Germany. The other reagents used to obtain antibiotic-loaded chitosan nanoparticles were of analytical grade of purity.

### 2.2. Preparation Methods

#### Nanoparticles’ Preparation

Chitosan nanoparticles (Chi) were obtained with ionic gelation method [31] using tri-polyphosphate (TPP) as cross-linking agent due to their counter ions which establish inter- and intra-molecular connections. This is a route to obtain nanoparticles based on the electrostatic interaction between compounds with opposite charges, with chitosan (cation) and TPP (anion) being the most frequently used. By adding TPP, drop by drop, to a solution containing chitosan, the polyanion binds to an amino group, which causes the polymer to suffer a gel ionization operation; then, nanoparticles were forms with centrifugation. The preparation of polymeric nanoparticles depends on the concentration of the polymer, its molecular weight, the chitosan/tri-polyphosphate ratio, the concentration of the bioactive principle, pH, the time and speed of stirring and centrifugation. This technique implies dissolution of chitosan polysaccharide in an aqueous acid solution (1% *v*/*v* glacial acetic acid) resulting in positively charged chitosan. Then, this solution is added to a TPP solution (0.1% *w*/*v* in ultrapure water) under stirring conditions and forms a positive–negative complex. 

For preparation of antibiotic-loaded polymer nanoparticles, metronidazole (in different concentrations) was added to chitosan solution, and adjustment of the pH of each product at 5.0 was performed by using solutions of 0.1 N HCl and 1 N NaOH. The as prepared nanocomposites were centrifuged at 9000 rpm for 1 h at room temperature. After this step, the supernatant was kept at 2–6 °C until further assessment. Finally, the antibiotic-loaded chitosan nanoparticles were lyophilized for 48 h and analyzed to achieve the aims of this study. The ionic gelation method requires cheap and easy-to-use materials and equipment, and having the advantage of the electrostatic interaction mechanism, instead of chemical synthesis, leads to the avoidance of possible toxicity of the reagents; however, its disadvantage is that nanoparticles are not produced on a large scale with a uniform size distribution. This technique is widely used because it has proven to be useful for obtaining various formulations of polymeric nanoparticles loaded with drugs. Each formulation code is presented in Table 1, which comprises the concentration of the drug, polymer and crosslinking agent.

After preparing the nanostructures, their characterization included particle size distribution (PSD), zeta potential, morphology (scanning electron microscope, SEM), encapsulation efficiency (EE), loading capacity (LC), Fourier transform infrared spectroscopy (FTIR) and spectrometry (UV-Vis). The characteristics of nanocomposites using these advanced techniques depended on the synthesis conditions.

### 2.3. Characterization Equipment

Characterization of the obtained samples was based on antibiotic-loaded polymer nanoparticles.

#### 2.3.1. Evaluation of Loading Capacity and Encapsulation Efficiency

Loading capacity and encapsulation efficiency of each composite was established using a Cary 60 UV-Vis spectrometer at 319.5 nm to measure the absorption of each supernatant after centrifugation. Loading capacity and encapsulation efficiency are expressed by the following formula:(1)$$\mathrm{Loading}\;\mathrm{capacity}\;\left(\%\right)=\frac{{W_{t}-W}_{f}}{W_{n}}\times 100$$
(2)$$\mathrm{Encapsulation}\;\mathrm{efficiency}\;\left(\%\right)=\frac{{W_{t}-W}_{f}}{W_{t}}\times 100$$
where

-*Wt* is the total content of metronidazole encapsulated initially into the polymer matrix;-*Wf* represents the amount of drugs in the supernatant;-*Wn* represents the weight of dried nanoparticles after lyophilization process.

Loading capacity is an indicator that reveals the amount of drugs that can be loaded into a quantity of nanocomposite; encapsulation efficiency is an index that shows the yield of the process of obtaining nanostructures.

##### Nanoparticles’ Size Assessment

Particles’ size distribution was established with DLS (dynamic light scattering) using a Zetasizer Nano ZS (Malvern, Germany), which evaluates the particle size, electrophoretic mobility and zeta potential. 

#### 2.3.2. Nanoparticles’ Morphology

Morphological features of metronidazole-loaded chitosan nanoparticles were analyzed using scanning electron microscope (Thermo Fisher Scientific, Waltham, MA, USA) equipped with an energy dispersive spectrometer (EDS, EDAX Octane Elite, Thermo Fisher Scientific, Waltham, MA, USA), which allows high-resolution morphological investigations of these types of nanocomposites. Nanostructured samples were dried at room temperature by spreading the suspensions on a glass plate; then, they were coated with gold under vacuum before examination. 

FTIR (Fourier transform infrared spectroscopy) characterization implied scanning of KBr tablets (in the spectral range of 4000 to 400 cm^−1^) that were obtained by compressing a mixture containing small amounts of KBr and antibiotic-loaded polymer nanoparticles. 

Release profiles were established at 37 °C in a horizontal shaker containing 50 mg/200 mL of drug-loaded chitosan nanoarchitectures by applying a few horizontal strikes. Dissolution media comprised phosphate-buffered saline solution pH 7.5, phosphate-buffered solution pH 7.0 and 0.1 N HCl solution pH 1.5. Metronidazole release operation was performed for 24 h and, at fixed intervals, 10 mL of sample was withdrawn and substituted with dissolution media then analyzed with UV-Vis spectrophotometer, at specific wavelengths to each dissolution media.

#### 2.3.3. Cell Viability Assay

HeLa cells were used for testing cytotoxic effect of metronidazole-encapsulated chitosan nanoparticles. After 24 h of incubation, in presence of the tested compounds, cell viability was evaluated with MTT assay. HeLa cell line was maintained in DMEM (Dulbecco’s Modified Eagle Medium, Biochrom AG, Berlin, Germany), and supplemented with 10% FSB (fetal bovine serum), 100 IU/mL penicillin and 100 µg/mL streptomycin at 37 °C in a humidified atmosphere of 5% CO_2_ in air. Evaluation of viability was based on MTT assay. HeLa cells were seeded in 96-well plates (density of 5 × 10^3^ cells/well), allowed to attach and grow overnight. Treatment with the polymeric nanoparticles anticipated the replacement of growth medium with new complete medium containing the NPs in doses ranging from 100 µg/mL to 0.01 µg/mL. After 24 h of treatment, the cells were washed and covered with 100 μL of fresh DMEM 10% FBS. An amount of 10 μL of MTT (5 mg/mL) was added into medium, and cells were incubated for another 3 h. DMSO (dimethyl sulfoxide, Merck, Darmstadt, Germany) was used to dissolve the formazan that was formed, and the absorbance was recorded at 570 nm. 

## 3. Results 

The loading capacity and encapsulation efficiency (Table 2) reveals that a higher antibiotic concentration determined a reduction in the loading capacity and an increase in the encapsulation efficiency.

The results of particle size and zeta potential evaluation showed that the concentration of the antibiotic did not intercede with the size and the positive zeta potential of the metronidazole-loaded chitosan nanoparticles.

Figure 3A shows the representative nanoparticles size distribution of chitosan nanoparticles and drug-loaded polymer nanoparticles for the ChiMet_2.2 formulation. The results revealed that ninety percent of chitosan nanoparticles had a size of 163.7 nm and metronidazole-loaded chitosan nanoparticles had a size of 201.3 nm. Thus, this confirmed that a small increase occurs when the drug molecules are encapsulated into the polymer structure. The zeta potential (Figure 3B) is based on the electrophoretic scattering of light for molecules, particles and surfaces in the size range of 0.3 nm–10 µm. The electrokinetic potential (zeta) represents the difference between the electric charges on the surface of the solid nanoparticles in the dispersing medium and the charges of the diffused electric layer. It also reveals the degree of repulsion of particles with the same charge. In the case of the new prepared formulations, a higher zeta potential indicated the stability of the dispersions that do not allow the aggregation of polymer nanoparticles loaded with antibiotics.

SEM micrographs, Figure 4A,B of the ChiMet_2.2 sample, show varied and dense nanoparticles, which are spherical in shape and have a porous chitosan matrix texture.

By comparing the morphology of antibiotic-loaded polymeric nanoparticles and non-loaded nanoparticles, it was observed that loading the drug into the chitosan structure did not significantly change the textural properties of the nanoparticles.

### 3.1. FTIR Characterization

Figure 5 displays the IR spectrum of the drug, polymer and metronidazole-encapsulated chitosan nanoparticles (ChiMet_2.2 sample). Pure metronidazole revealed characteristic peaks at 270 cm^−1^ attributed to the C-O stretching vibration, a N-O stretching at 1370 cm^−1^, a C-N stretching vibration at 1540 cm^−1^ and a O-H bond at 3230 cm^−1^. The IR spectrum of chitosan showed characteristic peaks at 3450 cm^−1^ assigned to the –OH and NH_2_ stretching vibration, and peaks at 1650 cm^−1^ attributed to the amide group.

IR spectrum of ChiMet_2.2 sample presented peaks at 1670 cm^−1^ corresponding to the amide group in chitosan, and peaks at 1490 cm^−1^ attributed to the –NO_3_ group in the metronidazole molecule and peaks at 1420 cm^−1^ assigned to the intermolecular hydrogen stretching vibration. 

From the analysis of the IR spectra, it can be assumed that the efficiency of the incorporated active principle depends on the technique of loading the drug into the polymer structure. 

### 3.2. Drug Release Profile

The in vitro release profiles of the antibiotic from the polymeric nanoparticles in different dissolution media (Figure 6A,B): phosphate-buffered saline (pH 7.5), phosphate buffer (pH 7.0) and 0.1 N HCl (pH 1.5). Up to one hour, there was an initial rapid release of about 30% of metronidazole, probably caused by the dissolution of the metronidazole crystals located on the surface of the chitosan nanoparticles. In the next 24 h, a prolonged release followed, possibly due to the diffusion of the antibiotic through the polymer matrix.

The release of metronidazole from the polymeric nanostructures shows that these drug carriers allow sustained and prolonged release of the drug at the target site, increasing the bioavailability of the drug and minimizing its adverse effects.

The release profile of metronidazole with different concentrations in the formulation showed that the release rate of the drug tended to increase as the amount of the drug in the sample increased, which lead to differences in the diffusion mechanism.

### 3.3. Cell Viability Assay

Cellular viability (%) (Figure 7) was calculated according to the following formula: % cell viability = [absorbance]sample/[absorbance]control × 100. Investigating the action of different concentrations of the ChiMet_2.2 sample on the viability of HeLa cell cultures revealed a moderate cytotoxic impact of these compounds, with cell viability reductions between 36.8% and 23.3%.

No significant differences were recorded between the different concentrations used in terms of the cytotoxic impact. However, even if the value of the cytotoxic impact is around 30%, from the point of view of the antitumor response, it has no therapeutic significance.

## 4. Discussion

The continuous development and improvement of drug delivery nanosystems is of major importance for the technologies used in the pharmaceutical field. Thus, obtaining nanostructures that act as a matrix for different active principles is significant due to the stability and control of drug release. Recently, the use of drug delivery systems in nanomedicine enhanced prevention, diagnosis and treatment of some threatening diseases by targeting the affected site while minimizing adverse effects. By using the ionic gelation method, bioactive molecules can be encapsulated in the polymer matrix to improve their effectiveness. Changing the amount of the drug in nanocomposites determined a loading capacity in the range of 70–62% and an entrapment efficiency in the range of 26–35%. The encapsulation capacity variation occurs due to the fact that chitosan nanoparticles precipitate faster, inhibiting the incorporation of the drug, which is followed by an increase in the diffusion of the drug when its amount in the formulation increases. Particle size distribution and zeta potential indicated no considerable modification when the antibiotic molecule was encapsulated into the polymer structure. The method used for the preparation of chitosan nanoparticles loaded with metronidazole led to a narrow size distribution of the nanocomposites. The complexity of the formation mechanism of the new nanostructures is probably due to the presence of phenomena, such as nucleation, growth, self-assembly and aggregation, that take place at the same time, thus determining the characteristics of the newly prepared formulations. If the predominant phenomenon is nucleation, a larger number of nanoparticles with smaller sizes are obtained; if it is molecular growth, self-assembly or aggregation predominates, and larger nanoparticles are formed. The conditions predetermined with the preparation method of the new compounds determine the predominant phenomenon, with them being the factors that influence the morphology and size distribution of the obtained nanoarchitectures. It was observed that an increase in the amount of the drug in the formulation implies a slight increase in the size of chitosan nanoparticles loaded with metronidazole from 163.7 nm to 201.3 nm, thus suggesting that the incorporation of the active molecule in the polymer matrix does not significantly change the size of the nanostructure. Morphology analysis performed with scanning electron microscopy indicated a porous surface of pure chitosan nanoparticles and spherical-shaped particles of the tested metronidazole-immobilized chitosan nanoparticles. A difference in the degree of agglomeration in nanocomposites compared to chitosan nanoparticles can also be observed. The IR spectra were recorded at the wavenumber range of 4000–400 cm^−1^ to evaluate chemical interactions between metronidazole and polymer molecules. FTIR spectral investigations, performed to analyze the alteration of chemical structure, revealed no significant physicochemical interaction between metronidazole and chitosan, concluding that these compounds are compatible with each other. The small changes in the absorption spectra of the new formulation of chitosan nanoparticles loaded with metronidazole possibly occurred due to the antibiotic molecule that changes the functional groups in the polymer. The dissolution assay indicated that the antibiotic was immobilized in the polymer structure due to the crosslinked gelation technique. The release profile suggests that the active substance was released in a controlled, prolonged manner from the polymer matrix with a non-Fickian behavior, indicating a transport of the active principle via diffusion, which is associated with polymer relaxation. Changing the metronidazole concentration did not significantly change the release profile of the drug from the tested samples and reached 80% for a 24 h period. The swelling and relaxation of the polymer could have a significant role in the release mechanism of the embedded substance. This could be due to the rapid dissolution of the drug on the surface of the chitosan nanoparticles that created pores, and its release from the polymeric nanostructure would have occurred through those pores. Towards the end of the delivery time, a decrease in the release speed could be observed due to the increase in the diffusion path of the drug. MTT testing is a quantitative method used on a large scale to determine, in vitro, the cytotoxic effects of some products (biomaterials, drugs, hybrid compounds) on cell lines. The cytotoxicity test was used to evaluate the possibility of the new formulation based on chitosan nanoparticles loaded with metronidazole having a therapeutic action without toxic effects on the cells. These findings presume promising applications of drug carriers in pharmacological and medical areas. Nowadays, there is a particular interest in obtaining polymeric nanoparticles through the ionic gelation method, which can be later loaded with new active ingredients and used both to prevent and to treat serious infections caused by various pathogens. This research aimed to highlight the possibility of expanding the field of applications of drug carriers based on polymeric nanoparticles loaded with different active principles to prevent and treat dangerous infections but also to reduce the side effects of drugs. An important aspect to consider refers to the chemical structure of the active molecule; therefore, the chemical affinity between the polymer and the drug can reveal surface interactions. Considering this factor in the design of new formulations could lead to the functionalization of the surface of polymeric nanoparticles to improve their performance as controlled drug release systems. This work aimed to obtain a new formulation of polymeric nanoparticles loaded with an antibiotic in which the structural and morphological characteristics were evaluated according to the conditions and methods of preparation and thus obtaining nanoparticulate systems with prolonged release of the drug without significant cytotoxicity. Further research should focus on testing chitosan nanoparticles loaded with metronidazole in different medical fields, and is expected to obtain a special performance of the transport and release of the drug at the target site. The prepared nanoparticles were characterized for their use in the delivery of active principles in biological systems, focusing on new formulations that reduce the adverse effects of drugs.

## 5. Conclusions

This study presents the synthesis and characterization of metronidazole-loaded chitosan nanoparticles as carriers for the antibiotic target site. The ionic gelation technique was used to prepare polymer nanoparticles. Chitosan is a natural polymer obtained with the deacetylation of chitin in an alkaline environment which is then metabolized by a human enzyme such as lysozyme. It is a cheap and easy to obtain, biocompatible, biodegradable and non-toxic compound that acts as a matrix for the inclusion of active principles. They also have a prolonged release at the site of action, which is followed by the dissolution of the host. Structural and morphological properties of as prepared delivery systems were performed using FTIR, SEM techniques and UV-VIS spectroscopy in order to investigate the release behavior. FTIR and SEM analyses indicate that the loading of the polymer matrix with an active principle does not significantly change its structural and textural characteristics. Moreover, the prolonged and controlled release of the molecule helps modern medicine in the diagnosis, prevention and treatment of many diseases. The results of the particle size distribution indicated a slight increase from 163.7 nm, in the case of uncharged polymer nanoparticles, to 210.3 nm, in the case of chitosan nanoparticles loaded with metronidazole. The cytotoxicity profile revealed that the cytotoxic impact was insignificant. Based on the obtained results, drug carriers can be successfully used for enhancing the encapsulation and biodistribution of metronidazole at the target site. Chitosan possesses low toxicity, indicating that it can be used in the biomedical field as a drug delivery nanosystem, thus promising various formulations which enhance drug therapy and limit side effects. Incorporating the antibiotic into the structure of chitosan nanoparticles can offer multiple advantages in the creation of transport systems and the localization of the active substance. The characteristics of nanocarriers such as the shape, size, release profile and cytotoxicity are significant in the design of drug delivery systems. Moreover, a promising strategy to combat antibiotic-resistant microorganisms and to reduce nosocomial infections is to use drug-loaded nanocarriers.

## Figures and Tables

**Figure 1 jfb-14-00538-f001:**
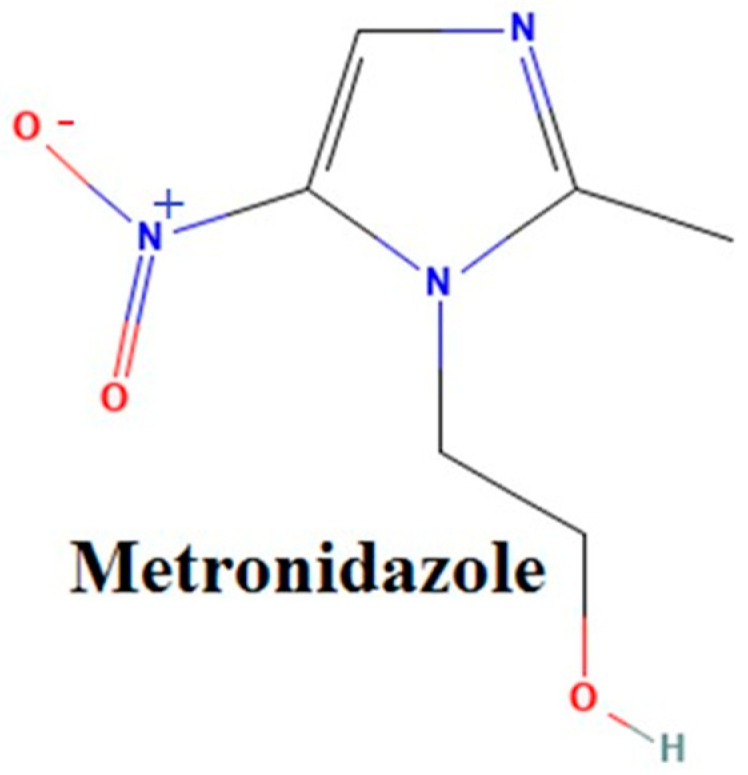
Chemical structure of metronidazole molecule.

**Figure 2 jfb-14-00538-f002:**
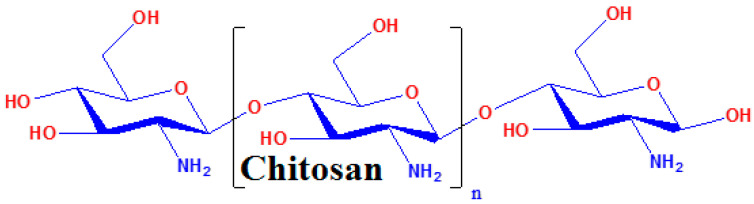
Structural representation of chitosan.

**Figure 3 jfb-14-00538-f003:**
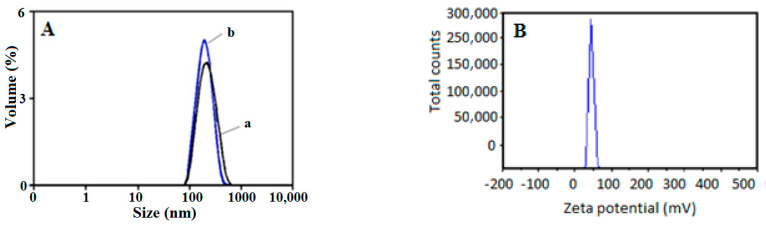
Nanoparticles’ size distribution representation: (**A**) chitosan particles (a) and metronidazole-loaded chitosan nanoparticles (b) and (**B**) zeta potential of chitosan-loaded metronidazole nanoparticles.

**Figure 4 jfb-14-00538-f004:**
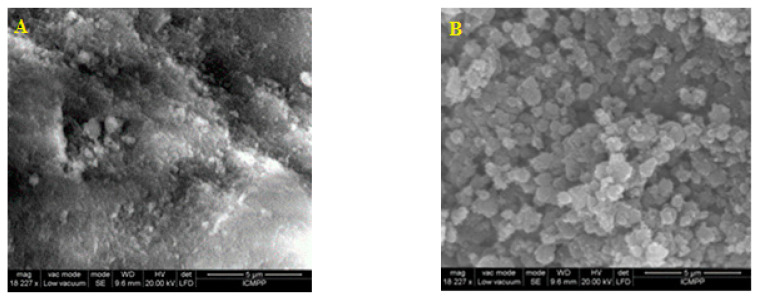
(**A**) SEM micrograph of chitosan. (**B**) SEM image of metronidazole-loaded chitosan nanoparticles.

**Figure 5 jfb-14-00538-f005:**
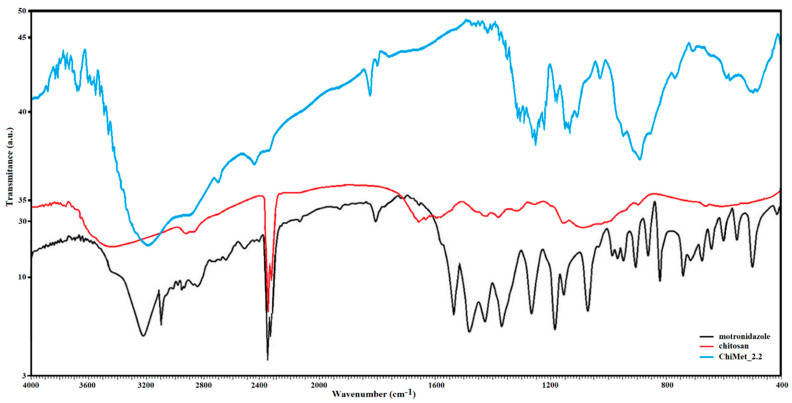
IR spectrum of pure metronidazole. IR spectrum of pure chitosan and IR spectrum of ChiMet_2.2 (metronidazole-loaded chitosan nanoparticles).

**Figure 6 jfb-14-00538-f006:**
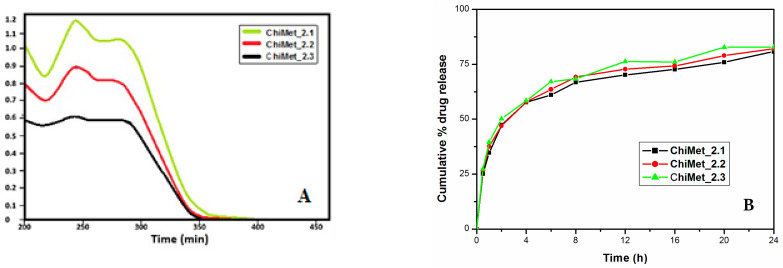
(**A**) UV–VIS spectrum of ChiMet_2.1, ChiMet_2.2, ChiMet_2.3. and (**B**) drug release profile of metronidazole-loaded chitosan nanoparticles.

**Figure 7 jfb-14-00538-f007:**
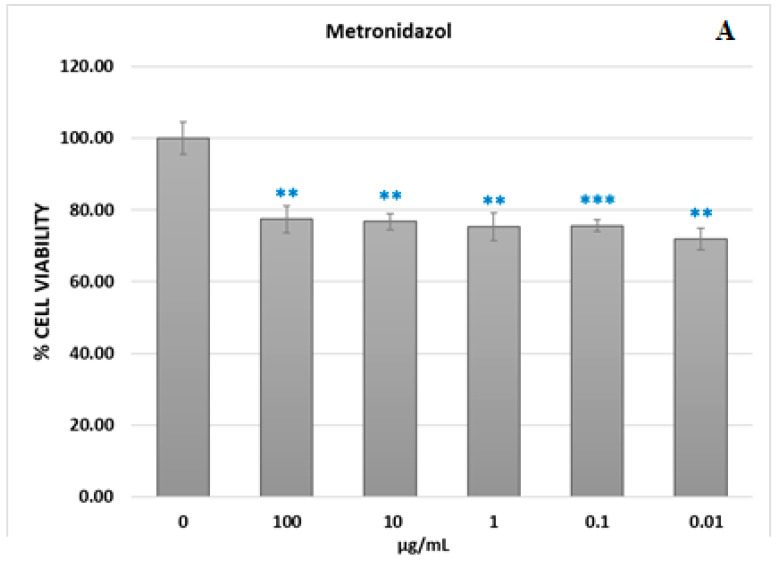
Viability of HeLa cells determined with MTT assay at 24 h after the treatment with chitosan-loaded metronidazol nanoparticles (**B**) or metronidazol NPs (**A**). Statistical significance was evaluated with paired *t* test, with significance thresholds of <0.05. ** = <0.01; *** = <0.001.

**Table 1 jfb-14-00538-t001:** The proportion of chitosan, TPP and metronidazole used in different formulations.

Code of Each Formulation	Chitosan (mg)	Crosslinking Agent, TPP (mg)	Metronidazole (mg)
Chi_1	100	50	-
Chi_2	150	50	-
Chi_3	200	50	-
ChiMet_2.1	150	50	100
ChiMet_2.2	150	50	150
ChiMet_2.3	150	50	200

**Table 2 jfb-14-00538-t002:** Loading capacity and encapsulation efficiency of the noncompounds.

Sample Code	Metronidazole/Chitosan Conc.	Loading Capacity, %	Encapsulation Efficiency, %
ChiMet_2.1	100/150	70	26
ChiMet_2.2	150/150	67	30
ChiMet_2.3	200/150	62	35

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
