# Peer review of "Development and Functionalization of a Novel Chitosan-Based Nanosystem for Enhanced Drug Delivery"

_jfb, 2023, doi:10.3390/jfb14110538_

Round 1

Reviewer 1 Report

Comments and Suggestions for Authors

The present manuscript entitled “Preparation, physicochemical and genotoxicity assessment of polymer composite as nano-carriers for antibiotic molecules” by Grierosu et al., describes the metronidazole encapsulated chitosan nanoparticles using the ion gelation route and evaluates their properties. Furthermore, scanning electron microscopy revealed a narrow particle size distribution and IR spectra revealed characteristic peaks of the drug and polymer nanoparticles and also Genotoxicity assessment revealed that samples have no significant impact on tested cells. The authors report an interesting work. The objective and justification of the work are clear and the results discussion part is elaborated in a reasonable way. Therefore, I recommend it for publication. However, some major issues are detailed below which need to be addressed before its final acceptance in the Journal of Functional Biomaterials.

I advise the authors to take the following points into account while revising their manuscript.

Comment 1: The whole manuscript must be cross-checked thoroughly for English editing, grammatical, spelling mistakes, and syntax errors. So, I suggest the author's English language should be polished. So many superscript and subscript errors are there in the manuscript For e.g. Line 211, 4000 to 400 cm-1 should be 4000 to 400 cm-1; Line 237, 163,7nm should be 163.7 nm; Line 238, 201,3nm should be 201.3 nm; Line 254, 270 cm-1 should be 270 cm-1; 1370 cm-1 should be 1370 cm-1;  Line 255, 1540 cm-1 should be 1540 cm-1; 3230 cm-1 should be 3230 cm-1;  Line 257, NH2 should be NH2; So cross check in the whole manuscript correct the errors.

Comment 2: The Abstract needs to be revised, let the author focus main points and explain the research question clearly, and also mention the zeta potential in the abstract and narrow particle size distribution value in the abstract section.

Comment 3: In the introduction, the section looks like little bit lengthy, so revise it

Comment 4: Include the graphical abstract in the revised manuscript to attain a broad readership.

Comment 5: Chemicals procured details such as purity, manufacturer, place of origin etc., in the Materials and Methods Section. Also, mention the Metronidazole procured complete details and also mention all the chemicals used in the current study in the materials section in the revised manuscript text.

Comment 6: Figures 4(a) and 4(b) are not cited in the manuscript text. So it needs to be cited in the appropriate place and also SEM results discussion needs to be elaborated.

Comment 7: The SEM images should be clearer; please consider updating or replacing them to improve their quality.

Comment 8: Figure 5 looks stretched so redraw the figure by using Origin or Excel and provide the high-resolution Figure also mention the peak positions in the Figure. Also, check the x-axis title of Figures 3 and 4, should be Wavenumber (cm-1) instead of Wavelength (cm-1). So check and correct it.

Comment 9: Figure 7 is incorrectly numbered in the manuscript as (Figure 57). So correct it as (Figure 7).

Comment 10: UV-Visible analysis was mentioned in the manuscript. However, I did not observe any UV-Vis spectrum, so provide it in the Revised manuscript text.

Comment 11: Shift author contribution details from top to bottom of the manuscript after the conclusion section.

Comment 12: Regarding the conclusions section, include clear quantitative findings and more emphasis on the findings and its implication may be mentioned in the conclusion section.

Comment 13: The homogeneity of the reference section needs to be maintained. Journal names are written some in full form and some in abbreviation form. So Format the references accordingly to the journal's instructions.

Comments on the Quality of English Language

Moderate editing of English language required to the manuscript.

Author Response

The present manuscript entitled “Preparation, physicochemical and genotoxicity assessment of polymer composite as nano-carriers for antibiotic molecules” by Grierosu et al., describes the metronidazole encapsulated chitosan nanoparticles using the ion gelation route and evaluates their properties. Furthermore, scanning electron microscopy revealed a narrow particle size distribution and IR spectra revealed characteristic peaks of the drug and polymer nanoparticles and also Genotoxicity assessment revealed that samples have no significant impact on tested cells. The authors report an interesting work. The objective and justification of the work are clear and the results discussion part is elaborated in a reasonable way. Therefore, I recommend it for publication. However, some major issues are detailed below which need to be addressed before its final acceptance in the Journal of Functional Biomaterials.

I advise the authors to take the following points into account while revising their manuscript.

Comment 1: The whole manuscript must be cross-checked thoroughly for English editing, grammatical, spelling mistakes, and syntax errors. So, I suggest the author's English language should be polished. So many superscript and subscript errors are there in the manuscript For e.g. Line 211, 4000 to 400 cm-1 should be 4000 to 400 cm-1; Line 237, 163,7nm should be 163.7 nm; Line 238, 201,3nm should be 201.3 nm; Line 254, 270 cm-1 should be 270 cm-1; 1370 cm-1 should be 1370 cm-1 Line 255, 1540 cm-1 should be 1540 cm-13230 cm-1 should be 3230 cm-1;  Line 257, NH2 should be NH2; So cross check in the whole manuscript correct the errors.

We corrected the manuscript as you suggested.

Comment 2: The Abstract needs to be revised, let the author focus main points and explain the research question clearly, and also mention the zeta potential in the abstract and narrow particle size distribution value in the abstract section.

We revised the abstract as you suggested.

Comment 3: In the introduction, the section looks like little bit lengthy, so revise it

We appreciate your suggestion regarding the introduction, but the presented aspects seemed relevant to this topic and shortening it would reduce the number of words below 4000, which would lead to non-compliance with the journal's criteria.

Comment 4: Include the graphical abstract in the revised manuscript to attain a broad readership.

Comment 5: Chemicals procured details such as purity, manufacturer, place of origin etc., in the Materials and Methods Section. Also, mention the Metronidazole procured complete details and also mention all the chemicals used in the current study in the materials section in the revised manuscript text.

 We completed the materials section as you suggested.

Comment 6: Figures 4(a) and 4(b) are not cited in the manuscript text. So it needs to be cited in the appropriate place and also SEM results discussion needs to be elaborated.

We cited figures 4(a) and 4(b).

Comment 7: The SEM images should be clearer; please consider updating or replacing them to improve their quality.

We do not have clearer SEM images

Comment 8: Figure 5 looks stretched so redraw the figure by using Origin or Excel and provide the high-resolution Figure also mention the peak positions in the Figure. Also, check the x-axis title of Figures 3 and 4, should be Wavenumber (cm-1instead of Wavelength (cm-1). So check and correct it.

We corrected x axis

Comment 9: Figure 7 is incorrectly numbered in the manuscript as (Figure 57). So correct it as (Figure 7).

We corrected as fig 7.

Comment 10: UV-Visible analysis was mentioned in the manuscript. However, I did not observe any UV-Vis spectrum, so provide it in the revised manuscript text.

We added UV-Vis spectrum to the manuscript.   

Comment 11: Shift author contribution details from top to bottom of the manuscript after the conclusion section.

We shifted author contribution details from top to bottom of the manuscript after the conclusion section.

Comment 12: Regarding the conclusions section, include clear quantitative findings and more emphasis on the findings and its implication may be mentioned in the conclusion section.

We included quantitative findings in the Conclusion section.

Comment 13: The homogeneity of the reference section needs to be maintained. Journal names are written some in full form and some in abbreviation form. So Format the references accordingly to the journal's instructions.

We revised the Reference section according to journal's instructions.

Reviewer 2 Report

Comments and Suggestions for Authors

The concept of the article is not new

The authors incorrectly assessed the genotoxicity of the of metronidazole loaded chitosan nanoparticles using the MTT assay, which cannot assess genotoxicity. The MTT assay evaluates cell metabolic activity after exposure to a toxic agent, but not genotoxicity. 

For genotoxicity, analysis of the morphology of the cell nucleus (micronucleus, condensation of chromatin, etc.) is used. Analysis of oxidative damage - 8 oxoguanine, for example, or single or double strand DNA breaks (using antibodies), or RAPD-PCR.

SEM image quality is very low quality (Figure 4).

Additionally, it is necessary to performed the UV spectrum of metronidazole, chitosan and nanoparticles. Additionally, its necessary to evaluate the zeta potential.

Figure captions are very short and not informative.

The method of cell cultivation is not described.

Why was this cell type chosen?

The authors of the article presented the structural formulas of chitosan and metronidazole, but the scheme of conjugation and the formed nanoparticles was not presented.

English language articles are of very poor quality.

Comments on the Quality of English Language

It is necessary to build a proposal correctly

It is necessary to express thoughts more capaciously in one sentence

Author Response

Comments and Suggestions for Authors

The concept of the article is not new

The authors incorrectly assessed the genotoxicity of the of metronidazole loaded chitosan nanoparticles using the MTT assay, which cannot assess genotoxicity. The MTT assay evaluates cell metabolic activity after exposure to a toxic agent, but not genotoxicity. 

For genotoxicity, analysis of the morphology of the cell nucleus (micronucleus, condensation of chromatin, etc.) is used. Analysis of oxidative damage - 8 oxoguanine, for example, or single or double strand DNA breaks (using antibodies), or RAPD-PCR.

We revised the section that refers to the cell viability test.

SEM image quality is very low quality (Figure 4).

We do not have clearer SEM images.

Additionally, it is necessary to performed the UV spectrum of metronidazole, chitosan and nanoparticles. Additionally, its necessary to evaluate the zeta potential.

UV spectrum of metronidazole, chitosan and nanoparticles. and the zeta potential, respectively were added to the manuscript.

Figure captions are very short and not informative.

We revised figure caption.

The method of cell cultivation is not described.

We revised the section that refers to the cell viability test.

Why was this cell type chosen?

We revised the section that refers to the cell viability test.

The authors of the article presented the structural formulas of chitosan and metronidazole, but the scheme of conjugation and the formed nanoparticles was not presented.

English language articles are of very poor quality.

Comments on the Quality of English Language

It is necessary to build a proposal correctly

It is necessary to express thoughts more capaciously in one sentence

Reviewer 3 Report

Comments and Suggestions for Authors

The article entitled "Preparation, physicochemical and genotoxicity assessment of polymer composite as nano-carriers for antibiotic molecules" reported the development of a nano drug delivery system, based on chitosan nanoparticles loaded with metronidazole, as a potential targeting delivery metronidazole. The work is overall good and covers important area. However, there are some shortcomings that prevents its acceptance in the current form:

1- I suggest changing the title to: Development and Functionalization of a Novel Chitosan-Based Nanosystem for Enhanced Drug Delivery.

2- The experiment designs and methods needs to be mentioned and clearly presented in full description especially the characterization procedures.

3- The quality of figure 4 needs to be improved.

4- I suggest including this related important article in the references: 

https://doi.org/10.3390/polym14173697

Comments on the Quality of English Language

Moderate Editing is required.

Author Response

Comments and Suggestions for Authors

The article entitled "Preparation, physicochemical and genotoxicity assessment of polymer composite as nano-carriers for antibiotic molecules" reported the development of a nano drug delivery system, based on chitosan nanoparticles loaded with metronidazole, as a potential targeting delivery metronidazole. The work is overall good and covers important area. However, there are some shortcomings that prevents its acceptance in the current form:

1- I suggest changing the title to: Development and Functionalization of a Novel Chitosan-Based Nanosystem for Enhanced Drug Delivery.

 We changed the title as you suggested.

2- The experiment designs and methods needs to be mentioned and clearly presented in full description especially the characterization procedures.

 We highlighted some of these issues.

3- The quality of figure 4 needs to be improved.

 We do not have clearer SEM images.

4- I suggest including this related important article in the references: 

https://doi.org/10.3390/polym14173697

We included the article you suggested in the references.

Reviewer 4 Report

Comments and Suggestions for Authors

In the current manuscript entitled " Preparation, physicochemical and genotoxicity assessment of polymer composite as nano-carriers for antibiotic molecules", the authors obtained metronidazole encapsulated chitosan nanoparticles using ion gelation route and to evaluate their properties. Due to the advantages of the ionic gelation method. the synthesized polymeric nanoparticles can be applied in various fields, especially pharmaceutical and medical. The manuscript can be considered for publication after minor revision to increase the paper quality.

1.  Barring a few grammatical and typo errors, language is very good.
2.  Abbreviations must be defined when first used.

3. There is no need for incorporation Figure 1 and Figure 2 in the main text and they should be moved to supplementary data.

4. The resolution of FTIR figure is very low. It should be redrawn to be more readable

5. The FTIR images should be combined in one figure.

6. Zeta potentials for chitosan particles and metronidazole loaded chitosan nanoparticles should be performed.

7. The Metronidazole release should be performed at different pHs

8. The onset table in figure 9 is not clear

Author Response

Comments and Suggestions for Authors

In the current manuscript entitled " Preparation, physicochemical and genotoxicity assessment of polymer composite as nano-carriers for antibiotic molecules", the authors obtained metronidazole encapsulated chitosan nanoparticles using ion gelation route and to evaluate their properties. Due to the advantages of the ionic gelation method. the synthesized polymeric nanoparticles can be applied in various fields, especially pharmaceutical and medical. The manuscript can be considered for publication after minor revision to increase the paper quality.

  1. Barring a few grammatical and typo errors, language is very good.
    2.  Abbreviations must be defined when first used.
  2. There is no need for incorporation Figure 1 and Figure 2 in the main text and they should be moved to supplementary data.
  3. The resolution of FTIR figure is very low. It should be redrawn to be more readable
  4. The FTIR images should be combined in one figure.
  5. Zeta potentials for chitosan particles and metronidazole loaded chitosan nanoparticles should be performed.

We revised the manuscript as you suggested.

  1. The Metronidazole release should be performed at different pHs.
  2. The onset table in figure 9 is not clear

We revised this issue.

Round 2

Reviewer 1 Report

Comments and Suggestions for Authors

The authors revised their manuscript according to my majority of concerns. Still, I have some minor suggestions.

The Figures 3 & 4 caption was revised into (a) and (b), but in figure (a) and (b) was not mentioned so it need to be revised.

Figures 8 and 9 need to be categorized into (a) and (b) and revise their captions. 

Comments on the Quality of English Language

Minor editing of English language required.

Author Response

The authors revised their manuscript according to my majority of concerns. Still, I have some minor suggestions.

The Figures 3 & 4 caption was revised into (a) and (b), but in figure (a) and (b) was not mentioned so it need to be revised.

We revised this issue.

Figures 8 and 9 need to be categorized into (a) and (b) and revise their captions. 

We revised this issue.

Minor editing of English language required.

The minor editing of the English language was done.

Reviewer 2 Report

Comments and Suggestions for Authors

1. The authors claim that the release of metronidazole from the hydrogel occurs in 24 hours of incubation by 80%. At the same time, according to the data of the viability analysis after 24 hours (Figure 9b), it can be seen that there is no statistical difference between the different doses of the loaded drug. At the same time, the concentrations of metronidazole differ by 3-4 orders of magnitude (from 0.01 µg/ml to 100 µg/ml) and have the same cytotoxic effect. This discrepancy looks strange. Please discuss this point. 

2. It is also worth correcting the figure 9 itself: on the x axis, indicate the concentration of the substance in µg/mL. 

3. IR spectrums must be combined into one figure and not designated as separate figures. 

4. Lines 292-308. The methodological part of the MTT test should be moved to the materials and methods section

Author Response

  1. The authors claim that the release of metronidazole from the hydrogel occurs in 24 hours of incubation by 80%. At the same time, according to the data of the viability analysis after 24 hours (Figure 9b), it can be seen that there is no statistical difference between the different doses of the loaded drug. At the same time, the concentrations of metronidazole differ by 3-4 orders of magnitude (from 0.01 µg/ml to 100 µg/ml) and have the same cytotoxic effect. This discrepancy looks strange. Please discuss this point. 

In vitro testing on cell cultures didn't aimed to prove that metronidazole could be used as an anticancerous agent, but we investigated the biocompatibility on cells as metronidazole alone or incorporated into nanoparticles. According to DrugBank, LD50 in rats is about 5000 mg / kg, and EMEA reports, no indications were found about any vitro cytotoxicity effects. 

In figures the concentration of metronidazole is expressed as ug/mL. 

  1. It is also worth correcting the figure 9 itself: on the x axis, indicate the concentration of the substance in µg/mL. 

We corrected x axis as you suggested and fig 9 became fig 7.

  1. IR spectrums must be combined into one figure and not designated as separate figures. 

IR spectrums were combined into one figure as you suggested, fig 5.

  1. Lines 292-308. The methodological part of the MTT test should be moved to the materials and methods section

The methodological part of the MTT test was moved to the materials and methods section as you suggested.

Reviewer 3 Report

Comments and Suggestions for Authors

Thanks for addressing all comments successfully.

Comments on the Quality of English Language

Minor editing.

Author Response

Comments and Suggestions for Authors

Thanks for addressing all comments successfully.

We thank you very much for the suggestions.

Comments on the Quality of English Language

Minor editing.

The minor editing of the English language was done.
